# Modelling the visual world of a velvet worm

**Mikael Ljungholm***, **Dan-E. Nilsson**

Lund Vision Group, Department of Biology, Lund University, Lund, Sweden

* mikael.ljungholm@biol.lu.se

## Abstract

In many animal phyla, eyes are small and provide only low-resolution vision for general orientation in the environment. Because these primitive eyes rarely have a defined image plane, traditional visual-optics principles cannot be applied. To assess the functional capacity of such eyes we have developed modelling principles based on ray tracing in 3D reconstructions of eye morphology, where refraction on the way to the photoreceptors and absorption in the photopigment are calculated incrementally for ray bundles from all angles within the visual field. From the ray tracing, we calculate the complete angular acceptance function of each photoreceptor in the eye, revealing the visual acuity for all parts of the visual field. We then use this information to generate visual filters that can be applied to high resolution images or videos to convert them to accurate representations of the spatial information seen by the animal. The method is here applied to the 0.1 mm eyes of the velvet worm *Euperipatoides rowelli* (Onychophora). These eyes of these terrestrial invertebrates consist of a curved cornea covering an irregular but optically homogeneous lens directly joining a retina packed with photoreceptive rhabdoms. 3D reconstruction from histological sections revealed an asymmetric eye, where the retina is deeper in the forward-pointing direction. The calculated visual acuity also reveals performance differences across the visual field, with a maximum acuity of about 0.11 cycles/deg in the forward direction despite laterally pointing eyes. The results agree with previous behavioural measurements of visual acuity, and suggest that velvet worm vision is adequate for orientation and positioning within the habitat.

## Author summary

It is difficult to understand the roles that vision may have in animals with visual performance very different to our own. Many invertebrates such as flatworms, polychaetes, onychophorans, gastropod molluscs and numerous arthropods have tiny eyes with unknown visual abilities. At best, behavioural experiments can reveal visual performance limits for specific behaviours but they will not give general information about what is visible to the animal, which is crucial for understanding the roles vision may have. Here we use ray tracing applied to accurate anatomical/optical models of the eyes of a velvet worm to reconstruct the visual acuity in all parts of the visual field. We also use the calculated visual performance to closely simulate what animals would see in their natural habitat. The

**Data Availability Statement:** All relevant data are within the manuscript and its Supporting Information files.

**Funding:** M.L. and D.-E.N. were funded by the Knut och Alice Wallenberg foundation, grant to D.-E. N., KAW 2011.0062(https://kaw.wallenberg.org) and

the Swedish Research Council, grant to D.-E.N.,
grant no. 2015-04690 and Swedish Research
Council, grant to D.-E. N., grant no. 2019-04813
(https://www.vr.se). The funders had no role in
study design, data collection and analysis, decision
to publish, or preparation of the manuscript.

**Competing interests:** The authors have declared
that no competing interests exist.

method can be applied to any (preferably small) eye and offers an alternative strategy that
may yield information about the visual capacity that is otherwise hard to obtain.

## Introduction

A detailed mapping of an animal's visual performance across its entire visual field is extremely
informative for revealing the roles vision may have in that species [1–3]. Because a visual sys-
tem is energetically expensive, we can assume that its performance is matched to the needs of
the most demanding visually guided behaviour [4,5]. But visual performance typically varies
across the visual field and each visually guided behaviour has its own variation of performance
requirements across the retina [6,7]. As a consequence, behavioural assessments of spatial res-
olution, or other measures of visual performance, may be valid only for a specific behaviour,
under specific circumstances and at a specific part of the visual field. There is a risk for incor-
rect conclusions if such performance values are taken to be general and used to ask which roles
vision may have in a particular species.

To assess the different roles that vision may have in a species, it is necessary to know the
eye's performance in all parts of its visual field. Spectral sensitivities, signal-to-noise ratios, and
photoreceptor temporal properties, all determine what an animal can see, but perhaps the
most informative measures are the retinal sampling matrix (pixel density) and the angular sen-
sitivity of each photoreceptor. It is the latter types of information that determines the spatial
visual performance (visual acuity) and how it varies over the visual field. Such information can
be obtained by electrophysiological recordings from retinal photoreceptors, in combination
with optical techniques and anatomical data [1,3]. Mapping of ommatidial axes in compound
eyes or measurements of retinal ganglion cell densities in vertebrate eyes are examples of
approaches aimed at revealing the "pixel density" of vision.

In the many invertebrates with low resolution vision, it is a major challenge to obtain com-
parative electrophysiological measurements of the angular sensitivity of the photoreceptors
[8]. In contrast to arthropod compound eyes with pseudopupils or eye glow, the eyes of most
other invertebrates offer no obvious optical cues to their spatial sampling. For comparative
studies of vision, a viable approach is to accurately determine the optical geometry of the eye
and measure the refractive index of essential components. This information can be used for
3D ray-tracing to computationally reconstruct the angular sensitivity of each photoreceptor in
the entire eye. There are many commercial optical simulation programs available today. How-
ever, since these are usually designed for optical engineering, they require the optical elements
to be represented with analytical surfaces. This excludes accurate analysis of animal eyes that
often display asymmetries and complex shapes of optical components and photoreceptor
structures.

Our aim here was to develop a ray-tracing method that could be applied to a wide variety of
animal eyes. In addition to computationally reconstructing the eye's visual sampling, we also
generate a computational image filter for converting high resolution images or videos to views
as they would be seen by the animal eye. This allows for simulation of what the animal would
see as it moves through its natural habitat, and thus offers a powerful tool for unravelling the
visual ecology of a species. The method was developed to simply provide high accuracy model-
ling of vision with a computational efficiency needed for comparative studies. For this reason,
we decided against using a full wave optical approach, and instead use a method based on geo-
metrical optics.

Here we use the ray-tracing approach based on 3D anatomical information and optical data (refractive index and absorption) to accurately reconstruct the angular sensitivity of all photo-receptors in the eye of the onychophoran *Euperipatoides rowelli* (Fig 1). This is a few cm long terrestrial onychophoran found on the floor of Australian forests. It is known to visually locate dark refuges, but other visually guided behaviours are not yet known [9]. In this species, visual acuity was recently measured behaviourally [9], and it is of special interest to unravel the visual ecology of the poorly known velvet worms. Because velvet worms have rather small eyes (Fig 1), they are of potential interest for studies of low-resolution vision, which is a crucial evolutionary step towards more advanced visual roles [10].

## Results

### 3D optical model of the eye

Just like in other velvet worms, *E. rowelli* has a pair of small but prominent eyes located behind the antennae on the head (Fig 1). Light reflection in the cornea of live animals reveal a smooth corneal surface forming a near hemisphere. Internally the eyes contain a lens and a retinal volume surrounded by dark screening pigment (Fig 1). In a previous paper, the lens was shown to have a homogeneous and high refractive index (1.485), and both its external and internal curvatures contribute to focus light into the retinal volume [9]. The retinal volume is filled by rhabdom microvilli from the photoreceptor cells. There is thus no vitreous space separating the lens from the photoreceptor volume.

The photoreceptors are of microvillar type and extend from the back of the retina toward the lens (Fig 2). The receptors are packed in a mostly hexagonal pattern with 2–3 μm distance centre to centre. The microvilli extend from the core of the receptor roughly perpendicular towards the surrounding receptors and bundles of microvilli from neighbouring receptors partly interdigitate. The overall shape of the receptor is cone like (Fig 2).

A 3D reconstruction of the whole head reveals that the eyes only occupy a small part of the head (Fig 3A). The 3D-model of the eye's geometry (Fig 3B and 3C) is based on plastic serial-sections through a fixed and embedded eye. We made sections from 10 different fully-grown adults. Although the generally smooth shape of the eyes was very similar in all 10 individuals, some revealed small dimples or irregularities of the pigment layer in various locations. Because these small dimples occurred at seemingly random locations, and were not seen during dissection prior to fixation, we conclude that they are artefacts from fixation, embedding and sectioning, most likely resulting from different amounts of shrinkage of the retinal volume and

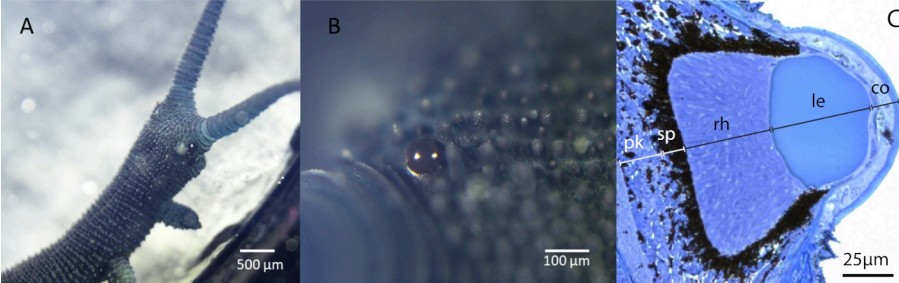

**Fig 1. Eye anatomy of velvet worms.** The eyes of *Euperipatoides rowelli* are small ocelli-like structures with a curved cornea, about 100 μm in diameter in adult animals. Scales: 500 μm in (A) and 100 μm in (B). (C) Light microscopic image of a stained semi-thin sections of the eye of *Euperipatoides rowelli*. Note that there is no space between the lens and the rhabdoms in the retina (photoreceptor volume). Cornea (co), lens (le), rhabdom layer/photoreceptor volume (rh), layer of screening pigment layer (sp), perikaryal layer (pk), adopted from [9].

**Fig 2. Photoreceptor ultrastructure.** (A) Transmission electron microscopy (TEM) image of cross sectioned photoreceptors with their cytoplasmic central cores (cc) separated by microvilli (mv). (B) TEM image of a section cut along the photoreceptor abutting the lens (le). The core extends from the back of the retina towards the lens. Microvilli protrude perpendicular from the core but often bend significantly and interdigitate with those of neighbouring receptors. (C) Schematic drawing of the structure of a photoreceptor showing its cone-like shape and the basal position of the screening pigment layer (sp).

the surrounding pigment layer. Rather than including these artifacts in an average model based on all 10 eyes, we selected a single eye that had a minimum of shape artefacts and made a 3D model based on this eye (Fig 4). Minor irregularities that were unique to this 'best eye' were evened out to agree with the general shape of the other eyes. The smooth but consistently asymmetric shape of the eye seen during dissection supports the above procedure. Furthermore, ray tracing in models with or without shape artifacts gave practically identical results, suggesting this is not an issue (S1 Fig). The 3D model reveals a distinctly asymmetrical retina with a posterior pouch (Figs 3B, 3C, 4A and 4B) corresponding to the forward part of the visual field. This asymmetry was consistently seen in all specimens also during dissection and is thus a real feature of the eye. In some sections, the corneal surface appeared to have a distinctly uneven curvature, but from the corneal reflection in live eyes (Fig 1) we conclude that the corneal surface is smooth and have a consistent curvature across the entire aperture. This suggests that the small irregularities seen in the corneal shape on some sections are artefacts of dehydration, embedding and sectioning, justifying a smooth curvature in the 3D model.

## Ray tracing

Based on the 3D model and refractive indices measured and assumed by Kirwan et al. [9], we performed 3D ray tracing to assess the optical performance of the eye.

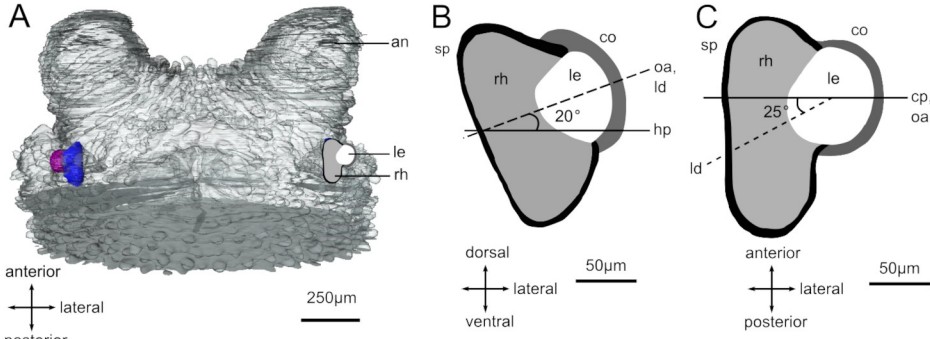

**Fig 3. Reconstructed head and eye of the E.rowelli.** (A) Orientation of the eyes in the head of the worm (reconstruction from sections of a single head, adopted from [9]. (B,C) The eye at higher magnification in vertical and horizontal planes. The elongated part of the retina extends to the rear in the head. Abbreviations: Cornea (co), lens (le), rhabdom layer (rh), screening pigment layer (sp), horizontal plane (hp), coronal plane (cp), antennae (an), Optical aperture (oa), lens direction (ld).

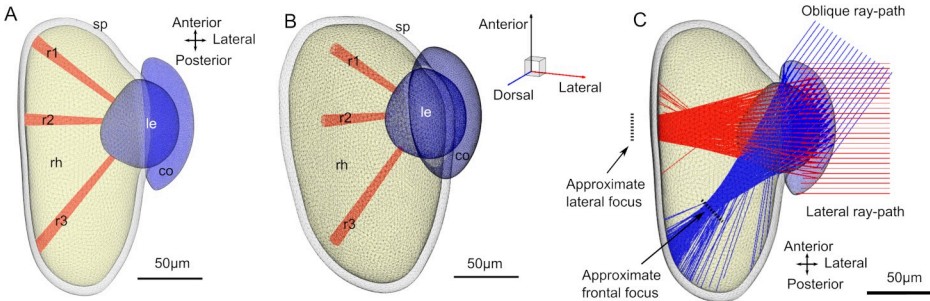

**Fig 4. 3D reconstruction of the eye and ray paths.** Each volume in the eye is limited by a triangular mesh surface. (A) Top down view of the eye. Each receptor is represented by a unique volume. Three selected receptors (r1, r2 and r3) are plotted to illustrate their size and position. (B) Orthographic side view of the eye to illustrate the 3D volume. (C) A reduced raytracing with two sources (only ~400rays) illustrates the difference of focal point for different incident angles. Rays from the lateral part of the visual field (red) are under-focused, whereas rays from the more frontal direction (blue) are focused inside the retina.

The immediate result from the ray tracing is the visualisation of how rays are focused (Fig 4). The eye has previously been concluded to be under-focused [9]. Our current ray-tracings reveal that this is only partly true. Rays entering from the frontal part of the visual field, where the retina is much deeper, are largely focused inside the retinal volume (Fig 4). Other parts of the retina are under-focused, with the centre being most severe and the posterior and ventral parts to a lesser degree.

By calculating the total absorption of all rays passing through each photoreceptor we could make a detailed reconstruction of the complete two-dimensional angular-sensitivity function. This is displayed for three different photoreceptors in Fig 5. The receptors are placed in different locations in the retina. Both shapes and sizes differ between locations. The higher resolution towards the edge of the visual field is partly due to the reduced effective aperture, which also implies a reduced absolute sensitivity. The smallest angular sensitivities were found in the forward direction, corresponding to the deep retinal pouch.

The angular sensitivities of all photoreceptors in the retina were used to generate a resolution map of the entire visual field, (Fig 6). The resolution differs markedly for horizontal and vertical image details. This astigmatism is particularly severe at the periphery of the visual field. The direction of highest resolution lies in the forward direction (approximately 45˚ from the aperture normal) for both the horizontal and vertical resolution. This acute zone in the frontal part of the visual field has a resolution of about 0.11 cycles/degree. Fig 6 also reveals the field of view of the eye, (using a criterion where the total absorption falls below 5% of peak value in the most peripheral photoreceptors). This information can be extracted without calculating the resolution. The total field of view is about 140˚ in the horizontal plane and 130˚ in the vertical plane. Additionally, Fig 6 also shows that because of the thicker retina, the total amount of absorbed light in the forward direction is higher than in the centre, despite the relatively smaller pupil area.

## Simulations of vision

To test the effect of the resolution across the visual field we generated an image filter that returns the modulation of an input image. This can be used to filter any image to reveal the spatial information the eye can pass to the nervous system. Using test images with a sinusoidal dot matrix (hexagonal array of 2D Bessel functions) close to the resolution limit, it is possible to illustrate the spatial performance across the visual field (Fig 7). The performance does not

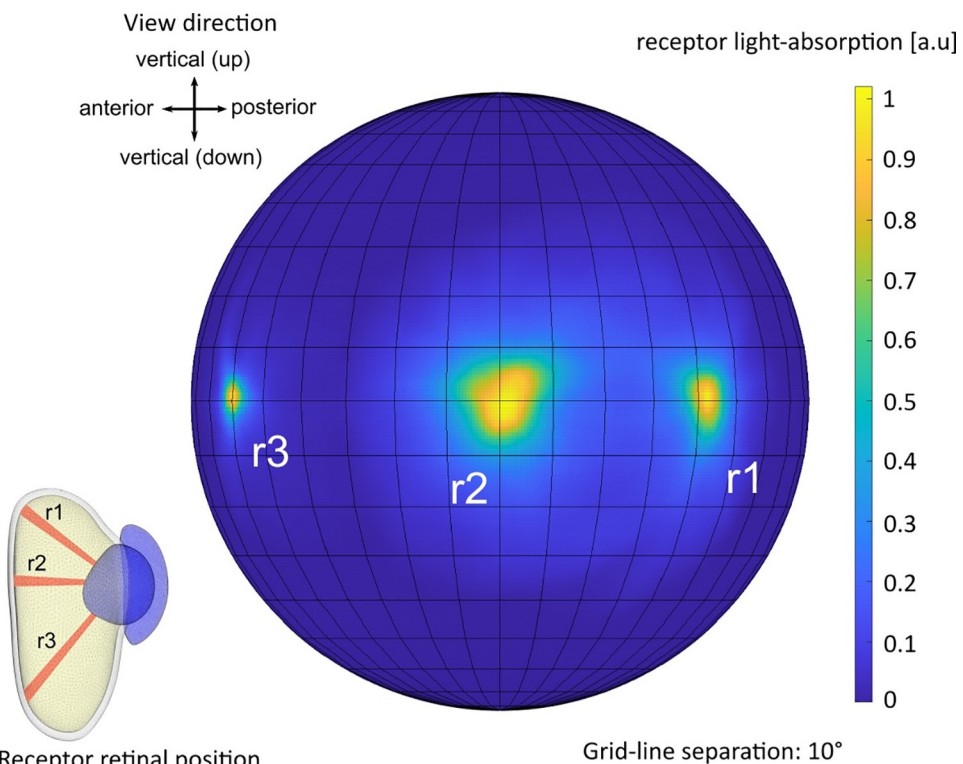

**Fig 5. Single receptor angular sensitivities.** Spherical orthographic projection heat maps of the angular sensitivity calculated for three individual photoreceptors. The receptor light-absorption scale is normalized separately for each receptor. The receptors r1, r2 and r3 are positioned in the front, centre and back of the retina respectively.

vary dramatically across the visual field but some directions appears more blurred than others. An obvious result is also the low contrast even of spatial details much coarser than the resolution limit. This is a consequence of the large flanks on the angular sensitivity functions. Image details of 0,095 cycles/degree are reasonably resolved in the forward part of the visual field but not in the retinal centre (lateral direction).

We also simulated the test patterns used for behavioural experiments by Kirwan et al. [9] (Fig 8). Here, the finest resolved pattern generates a receptor modulation of 33%, which can be taken as the smallest difference in receptor signal that the eye could discriminate at the absolute light levels used in the behavioural tests. Test patterns that did not evoke a behavioural response also clearly produce extremely faint modulation across the retina.

It is of course of interest to test what the optics can pass of a more natural image. We recorded images from the natural habitat, taken from velvet worm vantage points towards structures under which the animals were found to hide during the day. Passing such images through the computed spatial filter (Fig 9) reveals that the eyes of *E.rowelli* are capable of resolving major structures important for finding refuges in the environment, or to find their way out of a refuge. Spatial details fine enough to see other velvet worms, predators or potential insect prey are clearly beyond the capability of velvet worm vision. The image filter can also be applied to video sequences, which provide a strong impression of what a velvet worm can see. There is no sexual dimorphism is eye size or geometry, but juvenile specimens with smaller eyes are obviously expected to perform worse than the adults modelled here. (https://lu.box.com/s/q5z22e69c992ucti6xd53inirkn6pn90. An example of a filtered video taken with an eqisolid angle fisheye lens in the velvet worms natural habitat in Australia. The filtered

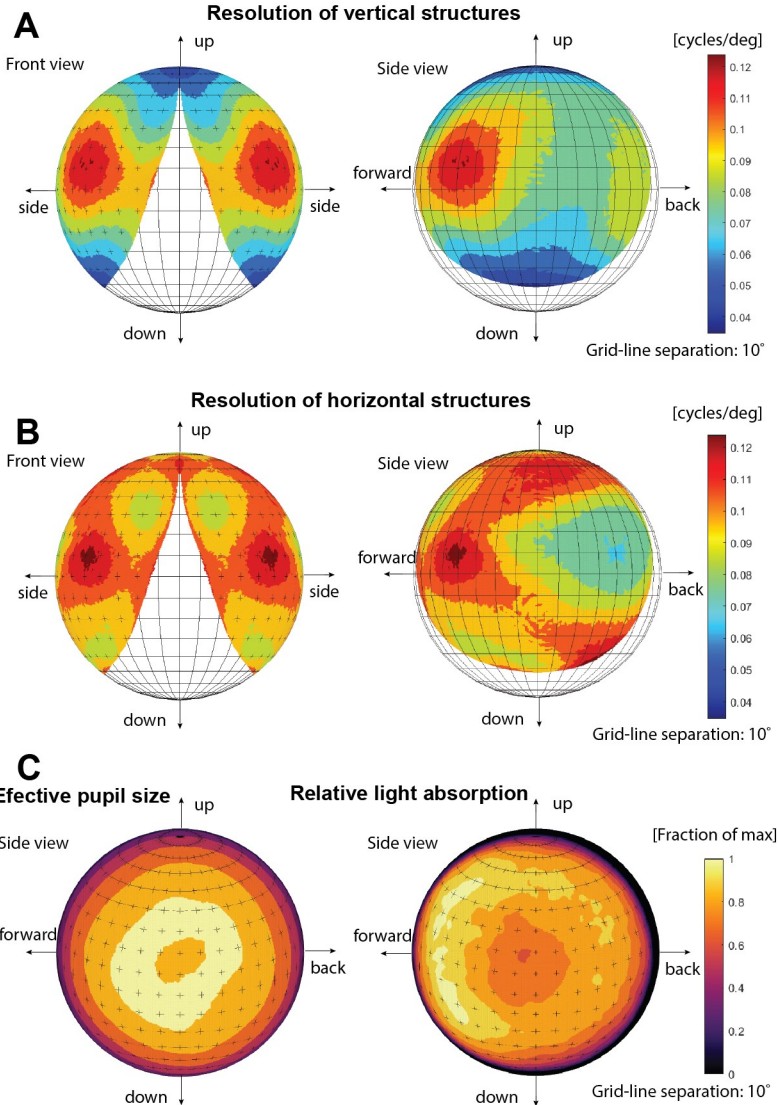

**Fig 6. Resolution in different parts of the visual field.** (A,B) Computed resolution (spatial cut-off frequency) for vertical and horizontal structures. The cut-off frequency was determined by Fourier transform of the individual photoreceptor sensitivity functions and determining the FWHM of the response in the vertical and horizontal direction in the Fourier plane. The field of view is displayed from both lateral and frontal directions. The visual field does not extend into the contralateral side, and consequently there is no binocular overlap between the two eyes. (C) Effective pupil area and computed relative light absorption by the retina in different viewing directions.

result is using the results of the raytracing absorption in individual photoreceptors, as it would be perceived by the receptors).

## Discussion

The ray tracing approach presented here for assessing visual performance and simulating vision in invertebrate eyes was successfully applied to the velvet worms. Similar approaches have previously been applied specifically to vision in box jellyfish [11–13], but here we have developed the method as a general tool for investigating vision in small eyes where there is no defined focal surface. These types of low-resolution eyes are common among invertebrates

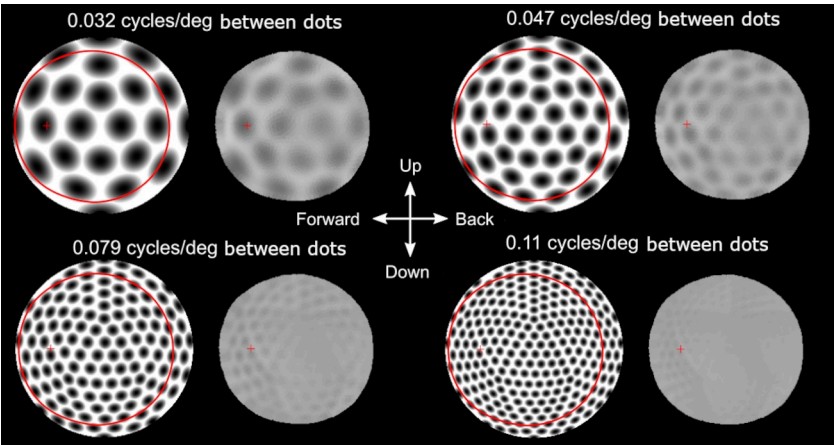

**Fig 7. Resolution of test images.** Input images and filtered images are shown for four different spatial frequencies (in each pair, the input image is to the left and the filtered output to the right). The smaller size of the filtered images reveals the limited visual field and is shown by the large red circle in the input image. The small red cross indicates the centre of the area with highest resolution. The input image is made up by 2d Bessel functions in a hexagonal grid on a sphere surrounding the eye so that the angular distance between each "dot" is roughly the equal to all near neighbours. It is this 3d image that is filtered by the eye. The stimuli are then projected on the 2d plane as a flattened 180° half sphere by an equisolid angle projection for visualization and the filtered signal is plotted with the same projection. The viewing directions for the right eye are indicated in the centre.

[10,14,15] but are notoriously difficult to study. Assessing what animals are able to see, and also what they cannot discriminate, is necessary for understanding visual ecology, i.e. how an animal can interact visually with its environment and the roles vision may have in different species [16,17]. Direct measurements of the image formed by the eye's optics has been a popular approach [18–20] in addition to behavioural measurements [9,12,13,21–25] and the traditional approach with intracellular recordings of angular sensitivity in individual photoreceptors [6]. A problem with behavioural approaches is that they target specific behaviours and not the general visual abilities of the animal. Electrophysiological approaches are direct measurements of spatial visual sampling, but unrealistically demanding for mapping visual sampling across the entire visual field. Especially for the many invertebrates with tiny eyes, the approach developed here is a superior and tractable way to gain insight into their visual ecology.

We based our geometrical model of the eye on histological sections. This may introduce artifacts from the histological preparation (shrinkage and deformations) that have to be taken into account. An emerging alternative is to use X-ray microtomography [20], which would significantly speed up the investigation. Many invertebrate eyes vary noticeably between individuals, raising questions on how this affects their visual performance. From the case investigated here, we found that even moderate modifications of the shape of the geometrical model have very minor consequences for the general visual performance. In an early attempt, we modelled vision in a single eye without removing even obvious shape artifacts (S1 Fig), but the results on modelled visual performance are practically absent. Apparently, in small eyes without defined focal planes and poorly focused optics, geometrical tolerances are rather large. This is in contrast to large high-resolution eyes where even very minor shape deformations will reduce the visual performance.

The method is based entirely on geometrical optics and ignores wave optical phenomena. But calculating the point spread function for green light ($\lambda$ = 500 nm) caused by diffraction in a $d$ = 100 μm aperture (1.22$\lambda$/$d$) yields 0.005 radians or about 0.3° (0.6° for a 50 μm aperture),

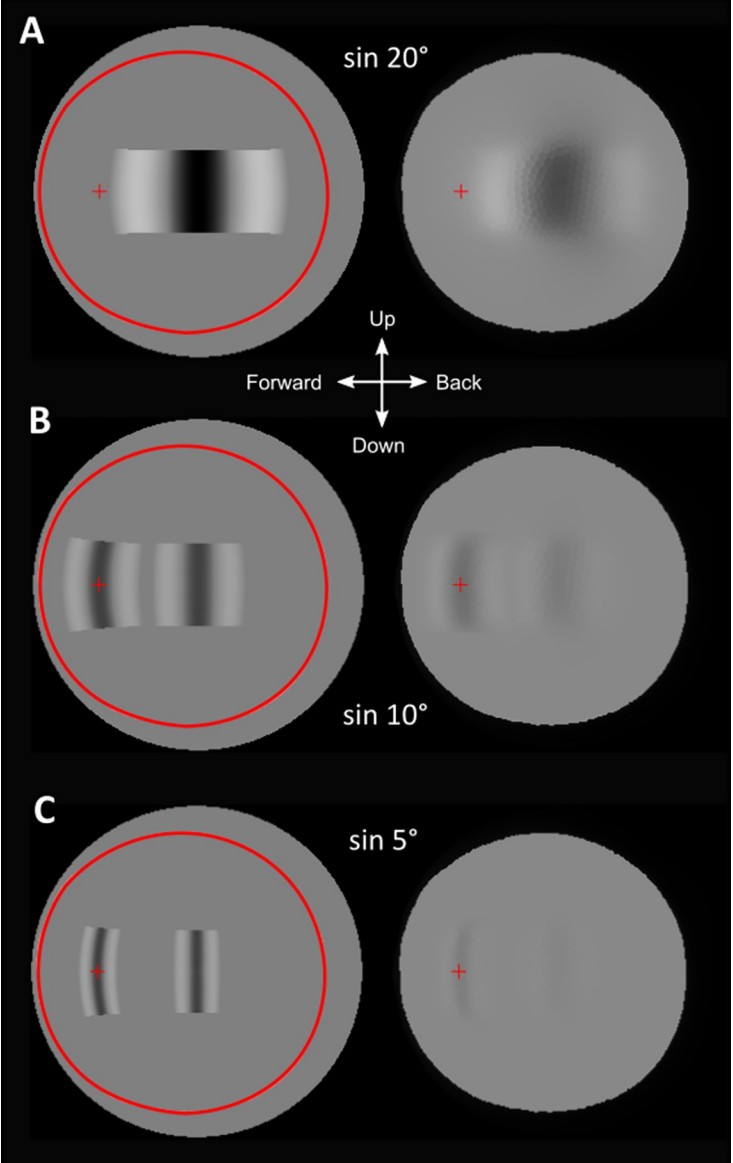

**Fig 8. Behavioural test patterns.** The input images (left) includes test patterns of piecewise sine functions or difference of gaussians used in behavioural experiments by Kirwan et al. [9]. The topmost figure shows a 20° target clearly visible even in the lowest resolving area. The centre and bottom are target patterns of 10° and 5° respectively in both the lateral part of the visual field and forward field. At 5° The pattern is barely visible in the forward direction (this is a pattern that did not evoke a behavioural response [9].

which can be safely ignored in a system where the smallest angular sensitivities are about 10°. This means that a full wave optics approach would not change the results, only vastly increase the computational task. It is also well known that diffraction is a limiting factor only large well focussed eyes in bright daylight [26]. Photon shot noise, on the other hand, is a main limitation to all other visual systems [26], and this can be directly calculated from the ray tracing as the absorbed fraction of light in individual photoreceptors if ambient radiances are known. Unfortunately, there is no data on the ambient light levels during which *E.rowelli* is active in their natural habitat. With images calibrated for photon flux, taken in the natural habitat, it would

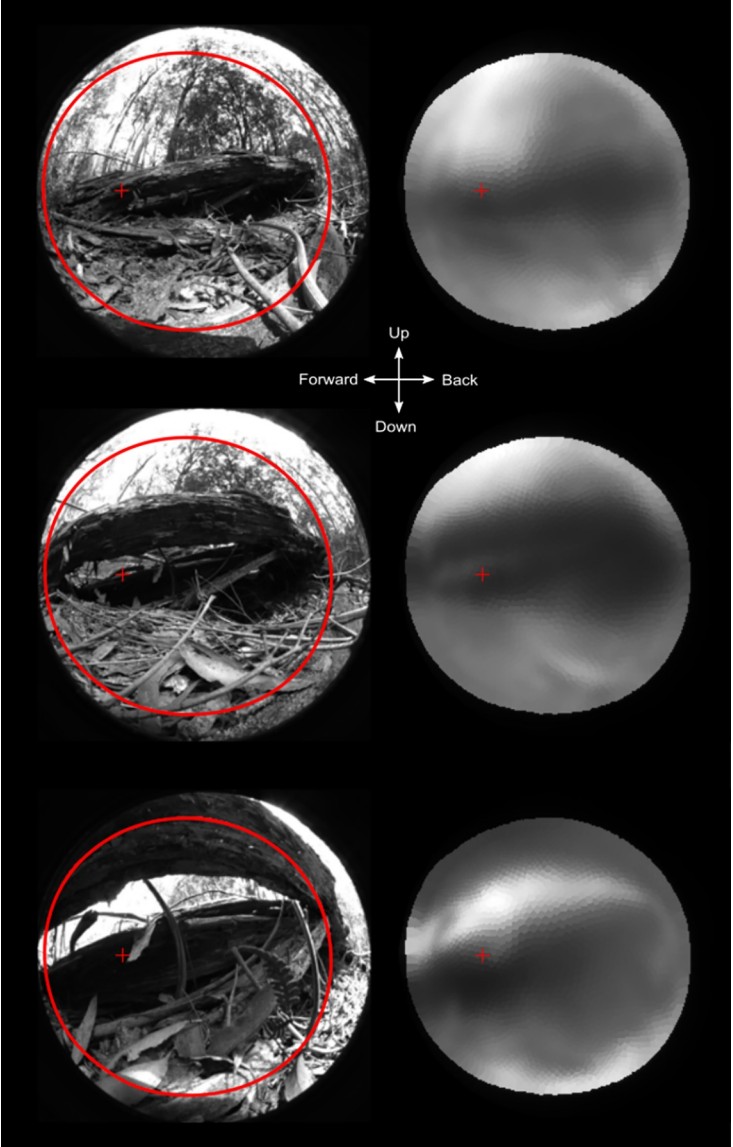

**Fig 9. Velvet worm view of their natural habitat.** The original images are 180˚ equisolid angle fisheye images taken from velvet worm vantage points in their natural habitat. All images contain major structures (fallen logs) under which velvet worms were found hiding during the day. Only these major structures are passed by the eye's optics. This would be enough to navigate towards dark hiding places but not for seeing other animals. The large red circle indicates the field of view and the small cross is the centre of the area with highest resolution. The images are for the right eye of the animal and the centre cross indicates the view directions.

be possible to also calculate photon noise and remove contrast that cannot be discriminated from the simulated images. Such an extension of the method would come very close to revealing exactly what an animal would be able to see in a given light intensity.

Another advantage offered by the ray-tracing approach is that the functional reason for any part of the eye's geometry or optics can be easily understood, and even measured by artificially modifying the geometrical/optical model of the eye. In principle, the method would be applicable to any kind of eye, including arthropod compound eyes and the high-resolution eyes of vertebrates and cephalopods. Especially for large camera-type eyes, the number of

photoreceptors are huge, which would call for an unrealistically huge computational power. This can be overcome by not simulating all photoreceptors but instead create patches of receptors in chosen places and extrapolating between them.

We of course cannot tell how animals experience their visual world, or even what they are capable of seeing without any knowledge on how the information is processed in the brain, but it is generally likely that eyes are not better than needed. Eyes are costly to build, maintain and run [5,27]. In terms of selection and adaptive traits, eye are thus an expense that has to be offset against its benefits, and their visual performance is likely to be fully exploited.

For the particular case of vision in velvet worms our results confirm behavioural measurements from previous studies [9]. These eyes provide only low-resolution vision, which is sufficient for general orientation in the habitat: trees, fallen logs and other large features can be distinguished. But the resolution is not good enough to detect or interact visually with other animals, be it prey, predators or conspecifics [10,9]. This corresponds to how the worm is observed to behave. It moves toward a prey but do not attack until it touches the prey with its antennae, assumingly using chemical senses to identify the prey. The limited spatial resolution is due to a retinal volume reaching all the way to the lens and the generally poor focusing. Much of the retina is under-focused, with exception of the posterior pouch responsible for forward vision. It is clear from the design of the eye, and perhaps not surprising, that spatial resolution in the forward direction is more important than in other directions.

One outcome of the revealed ray paths in the velvet worm eye is that the effective aperture is never as large as the lens diameter. This may be true for terrestrial invertebrate eyes in general and means that estimates of visual performance based solely on the external lens diameter may be significantly at error. Close to the edge of the visual field, the aperture is particularly restricted. This increases the maximum spatial frequency passed by the optics, but of course also reduces the light capture and thus makes the photon shot noise worse. If we assume that the eye is only just large enough to provide sufficient sensitivity for everyday use at dusk and dawn (the time when they exit from and enter their daily shelter) then the lower sensitivity at the periphery implies that its potentially higher spatial resolving power cannot be used. The situation is different for the deep retinal pouch looking in the forward direction. Here, loss of sensitivity by a smaller effective aperture in the periphery is compensated by the longer light path through the photoreceptor. This is also an obvious contributing reason for the retinal pouch. Increased resolution and sensitivity makes the retinal pouch into a foveal region where the spatial resolution is about twice that in the middle of the visual field (looking laterally), but with maintained light capture and thus uncompromised contrast sensitivity.

The ray-tracing method for assessing visual performance and simulating vision in eyes without defined focal planes has revealed previously unknown features of eye design in a velvet worm and provided a mechanistic explanation of earlier behavioural results [9]. The method was developed as a general tool, able to provide a full assessment of the spatial information in any type of eye, especially in all the small invertebrate eyes the visual capacity is otherwise hard to quantify and evaluate. The method allows the visual world of any animal to be uncovered in detail if the optical geometry and distribution of refractive indices can be determined.

## Methods

### Anatomy and optical model

The anatomical model was created by 3D reconstruction of histological sections from the head of a *E. rowelli*, using material from Kirwan et al. [9]. The samples were prepared in fixative (2.5% glutaraldehyde, 2% paraformaldehyde in 0.1 M sodium cacodylate buffer) for 12 hours, then washed several times with buffer and postfixed in 1% osmium tetroxide for 2 hours. The

samples were then again washed, dehydrated in an ethanol series, rinsed with acetone and pre-embedded in Epon overnight. The next day the samples were embedded in Epon and polymerized for 48 hours. The samples were cut into 3um thick sections with a RMX TPX microtome (Boeckeler Instruments, Inc., Tucson, AZ, USA) and stained with 1% toluidine blue in 1% borax. Images of the sections were taken with a 25x objective using a microscope camera (DS-Fi2-U3, Nikon Corporation, Tokyo, Japan, mounted on an Axiophot microscope, Carl Zeiss Microscopy GmbH, Jena, Germany) using imaging software NIS-Elements (version 4.20, Nikon Corporation, Tokyo, Japan).

A total of 10 eyes were initially sectioned and inspected for potential artefacts. It turned out that fixation and/or embedding frequently causes dimples or uneven shapes at random locations on the otherwise smooth cornea, lens and retinal pigment layer. To overcome the inaccuracies caused by such artefacts we selected the best eye, which was almost free from obvious artefacts. Comparison with the other sectioned eyes confirmed that the selected eye was representative. The general shape of the eye was consistent in all sectioned eyes, although there was some difference in size, reflecting the fact that all sacrificed animals were not of identical size. We decided against generating an average shape model from all the sectioned eyes because artefacts and size differences would obviously make it depart from the geometry of a live eye.

Reconstruction and segmentation of the digital 3D model was done using the Amira software (version 5.3.3, FEI Visualization Sciences Group, Burlington, MA, USA) and exported as. stl Delaunay surfaces. Minor local shape-irregularities that were unique to the selected 'best eye' were corrected in the 3D model by comparing with the microscopy images of the other eyes. These corrections made the model agree closely with the typical shape of all the eyes that were sectioned.

## Ray tracing

Assessment of the eye's optical performance was done in MATLAB using geometrical optics [28] and a ray-tracing approach. The light source was modelled as a parallel bundle consisting of 10000 evenly spaced rays in a circular beam with radius 50μm. The ray-tracing was repeated for 4000 different incident angles evenly distributed over a hemisphere covering the entire visual field. The surfaces of the geometrical model were made up of finite element mesh in the form of triangular surfaces. The surfaces where exported from AMIRA in the form of ".stl" files [29]. Each surface (cornea, lens and retina-wall) consisted of between 7000–17000 triangular surfaces. The surfaces where then partitioned using an Octree Space Partitioning (OSP) algorithm to reduce computation time. Each ray was traced to and in the octree volume using a parallelized version of Amanatides and Woo fast voxel traversal algorithm [30]. Intersection with each surface and ray was calculated using the Möller-Trumbore intersection algorithm [31–32], where refraction was calculated using Snell's law. The air surrounding the eye was given a refractive index of n = 1; cornea n = 1.4; lens n = 1.44 [9] and the retina was assumed a refractive index of n = 1.36, as commonly assumed for microvilli based retinas [33].

The photoreceptors were modelled as absorbing cones with a non-absorbing core (Fig 2), with their base at the back of the retina and pointing towards the lens (Fig 4). The photoreceptors where then placed in a roughly hexagonal pattern using AMIRA's meshing engine. The size of the hexagonal point mesh was created so that the average distance between two neighbouring centres where about 2.5 μm. This creates an overlap of the absorbing part of the receptors (microvilli) with the neighbouring receptors according to the retinal anatomy (Fig 2).

The absorption for the ray-path was calculated using Beer-Lamberts law with an absorption coefficient 0.0067 $\mu m^{-1}$ [6]. The angular sensitivity for each photoreceptor was calculated by the absorption from each source point interpolated over a cartesian angular grid (Fig 5). The

cut-off frequency for each receptor was then determined by the full width half maximum of the signal in Fourier space. The cut off frequency was then plotted onto a sphere (Fig 6) where the viewing direction for the photoreceptors was determined by the weighted centroid (above 30% signal) of the angle dependent absorption function for the receptor.

### Generation of image filter

The viewing direction and angular dependent absorption function where already calculated for each photoreceptor, these were used together with input images with a known image forming function to directly form a transfer matrix to filter images. In this case the camera used an equisolid angle fisheye projection. The projection was calibrated so that the spherical direction is known for each image pixel. The absorption profile for each receptor was then used to form a transfer matrix of the same size and angle calibration as the input image. This stack of transfer matrices were used to calculate the total absorption value in each receptor for a given input image. The output image was generated by creating a Voronoi cell map of the same size and angle calibration as the input image where the position of the Voronoi cells were determined by the viewing direction for each receptor. Each Voronoi cell was given a pixel value of the total absorption value for the corresponding receptor. An edge filter and a white filter was used on the output image to create the final output. The edge filter was created to cut off any cell that had a total absorption value below 5% of the maximum absorption. The white filter was created to compensate for the different absorption strengths of the receptors so that a uniform white input image would result in a uniform white output image.

## Supporting information

**S1 Fig. Spatial resolution using a 3d model with minimal alteration.** Computed resolution (spatial cut-off frequency) for horizontal structures using a 3D model with minimal altering after segmentation reconstruction. The cut off frequencies in the forward direction ~0.08 cycles/deg does not differ greatly from the optimized model ~0.1 cycles/deg. The cut-off frequency [cycles/deg] was determined by Fourier transform of the individual photoreceptor sensitivity functions and determining the FWHM of the response in the horizontal direction in the Fourier plane.
(TIF)

**S1 Code. The complete program for raytracing, absorption, analysis and filter generation.** The complete code used for raytracing and filter creation. This includes functions for importation and manipulation of surfaces and volumes, spatial partitioning, creation of photoreceptor approximation, ray source creation, raytracing, absorption calculation, absorption analysis, absorption result visualization, image filter creation and image filtering. Source code can be found at https://github.com/mLjungholm/Raytrace.git.
(ZIP)

## Acknowledgments

We are grateful to John Kirwan for providing results and material from [9], to Georg Mayer for providing animals and to Carina Rasmussen and Eva Landgren for preparing histological sections.

## Author Contributions

**Conceptualization:** Dan-E. Nilsson.

**Formal analysis:** Mikael Ljungholm.

**Funding acquisition:** Dan-E. Nilsson.

**Investigation:** Mikael Ljungholm.

**Methodology:** Mikael Ljungholm, Dan-E. Nilsson.

**Software:** Mikael Ljungholm.

**Supervision:** Dan-E. Nilsson.

**Visualization:** Mikael Ljungholm.

**Writing – original draft:** Mikael Ljungholm, Dan-E. Nilsson.

**Writing – review & editing:** Mikael Ljungholm, Dan-E. Nilsson.

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
