## [Decision Letter · Decision Letter 0]

3 May 2021

Dear  Mr Mikael Ljungholm,

Thank you very much for submitting your manuscript "Modelling the visual world of a velvet worm" for consideration at PLOS Computational Biology. As with all papers reviewed by the journal, your manuscript was reviewed by members of the editorial board and by several independent reviewers. The reviewers appreciated the attention to an important topic. Based on the reviews, we are likely to accept this manuscript for publication, providing that you modify the manuscript according to the review recommendations.

As you see all three referees praise the contents of the paper. One referee lists a number of suggested improvements which you should consider making, another raises some points to address and asks you to remove myriad typos and small errors that he has spotted but not itemised. Please read the paper carefully to find and remove them. The third has nothing but enthusiasm for your study.

Sincerely,

Tom Collett

Guest Editor

PLOS Computational Biology

Wolfgang Einhäuser

Deputy Editor

PLOS Computational Biology

[LINK]

As you see all three referees praise the contents of the paper. One referee lists a number of suggested improvements which you should consider making, another raises some points to address and asks you to remove myriad typos and small errors that he has spotted but not itemised. Please read the paper carefully to find and remove them. The third has nothing but enthusiasm for your study.

Reviewer's Responses to Questions

**Comments to the Authors:**

Reviewer #1: This is a valuable study of worm vision. A massive computational effort into the optics of the velvet worm eye shows that focusing is crummy, but good enough for achieving contrastful images. The applied methods as well as the obtained results are clearly presented, and so I have little to complain, except that I have seldom seen a paper so packed full of crazy typos and linguistic errors. I will only list for the abstract line 19 (where) and 21 (within). At each page there are 5-10 of similar stupidities. And of course, I started to curse when reaching line 287.

Apparently the authors do not check their work, as can be glanced also from their JEB (Ref. 10) paper. Should we become suspicious about possible carelessness in the calculations?

I regret to have to say this, because the paper’s results look very OK.

It may be worthwhile to reiterate that the lens is assumed to be homogeneous, as indicated by the previous optical studies. And, a definition of what is considered to be a rhabdom vs rhabdomeres (line 103) may be worthwhile. One example of other sloppiness: in Fig. 2, pg is pigment granules, in Fig. 4 screening pigment layer. Fig. 4 has no A and B, and so on. It would take me an hour or more to list all these scholarly shortcomings, but I do not regard that as the task of a referee.

Reviewer #2: This paper provides detailed results of an anatomical investigation into the eye morphology of a velvet worm and uses this information to construct a model of the resolution and sensitivity of the visual system. Such models have been challenging in this kind of eye in the past, due to the lack of a defined image plane. The current study uses ray tracing to uncover the influence of optical components of the eye, and photoreceptor angular sensitivities to determine the resolving power of the retina and combines these to produce a model of the visual scene as viewed through the velvet worm eye.

Overall, the paper is clearly structured and well written and presents novel data of wide interest to the scientific community. The computational aspects of visual modelling conducted in the study place it nicely in the remit of PLoS Computational Biology.

General comments:

Although covered nicely in the discussion, the introduction could do with more information on the behaviour and ecology of the velvet worm, and the various visual tasks that they are known to perform.

Line 98 and 128 – ‘Glare’ is perhaps the wrong word here. Consider using ‘reflection’ or ‘reflected light’ instead.

Line 98 – Is ‘lenses’ the correct word here? Do you mean ‘cornea’.

Line 99 – Please explain why the internal area of the eye is classified as a lens rather than an empty void of vitreous. Is it a hard structure?

The selection of the ‘best eye’ (line 125) was somewhat subjective. Is there a way you can make this selection process more robust? Are you confident that all the eyes were not subjected to the same fixation artefacts, causing similar deformation across all eyes, and therefore also in your ‘best eye’? Any additional information that improves the readers confidence in this eye selection process would be welcome.

Some of the figures would be better consolidated into multiple panels of a single figure. For example figure 1 and 2 would work well consolidated together. Also figures 5 and 6.

Minor suggestions

Please check the spelling of “were” and “where” throughout the manuscript.

Line 55 – Please remove the word ‘obvious’.

Line 74 – Missing full stop.

Line 78 – ‘…in addition to computationally restructuring the eye’s…’

Line 89 – “…behaviourally [10],…”

Line 92 – Consider removing “obviously”.

Line 130 – please remove ‘…indeed…’.

Line 144 – “…visualisation of how rays…”

Line 147-8 – Please be more specific than ‘more or less severely’.

Line 155 – Please be more specific than ‘obvious’. Perhaps state how different they are in viewing area?

Line 171-2 – consider replacing ‘absorption length’ with ‘retinal thickness’.

Line 177 – ‘…does not extend into the…’

Line 197 – Presumably X% is missing a value.

Line 204-206 – Last 3 sentences in the caption are repeating the methods so can be removed.

Line 210 – Check wording around ‘…reproducing major and structures…’.

Line 224 – Please remove ‘…the iconic…’.

Line 257 – Please briefly explain why you ‘did not do that here’.

Line 287 – ‘…but of course also…’

Figure 7 – Is ‘intensity’ the most appropriate label for the colourmap? This could easily cause confusion with light intensity.

Figure 8 – The label ‘Relative light absorption’ is a little misleading. Is this a measure of retinal thickness? Why not use ‘retinal thickness as your label, given that you are not directly measuring light absorption?

Figure 9 – It would be great to have an indication on the figure or in the caption of the angular size of the image. Presumably that the main image is a flattened hemisphere with an angular size of 180 degrees.

Reviewer #3: this is one of those rare manuscripts that is uniformly excellent. The methods appear rock solid and well executed, and the paper is very well written. The animals are of course not of regular interest to most biologists, but the methods provide a blueprint for investigating the vision of invertebrates with camera eyes, where RGCs or pseudopupils do not provide a guide, such as arachnids, molluscs, and certain cnidaria and annelids.

My only minor concern is that the eye morphology examined may not be as universal for the species as assumed. It is quite an asymmetric eye in an animal that does not appear to require high-resolution vision, so natural selection for exact eye shape may be quite relaxed. The authors do say they picked one eye out of ten as representative, but it would be worth seeing a panel of the other eyes to get a sense of variation. There may also be variation with respect to age, sex, and population, so I think the authors should be cautious about their conclusions.

**Have the authors made all data and (if applicable) computational code underlying the findings in their manuscript fully available?**

Reviewer #1: None

Reviewer #2: Yes

Reviewer #3: Yes

PLOS authors have the option to publish the peer review history of their article (what does this mean?). If published, this will include your full peer review and any attached files.

Reviewer #1: No

Reviewer #2: No

Reviewer #3: No

Figure Files:

Data Requirements:

Reproducibility:

References:

---

## [Editor Report · Decision Letter 1]

9 Jun 2021

Dear Mr Ljungholm,

Thank you for your revised version of 'Modelling the visual world of a velvet worm' and for following the reviewers' suggested changes. Since you have dealt with all the points that the reviewers raised, we are happy to say that your paper is provisionally accepted for publication in PLOS Computational Biology. It is fascinating that even the velvet worm has a means to prioritise the region of eye looking forwards.

Before your paper can be formally accepted you will need to complete some formatting changes, which you will receive in a follow up email. A member of our team will be in touch with a set of requests.

Best regards,

Tom Collett

Guest Editor

PLOS Computational Biology

Wolfgang Einhäuser

Deputy Editor

PLOS Computational Biology

---

## [Editor Report · Acceptance letter]

15 Jul 2021

PCOMPBIOL-D-20-02002R1 

Modelling the visual world of a velvet worm

Dear Dr Ljungholm,

I am pleased to inform you that your manuscript has been formally accepted for publication in PLOS Computational Biology. Your manuscript is now with our production department and you will be notified of the publication date in due course.

With kind regards,

Katalin Szabo
